# Alcohol Sensor Based on Surface Plasmon Resonance of ZnO Nanoflowers/Au Structure

**DOI:** 10.3390/ma15010189

**Published:** 2021-12-27

**Authors:** Haowen Xu, Yutong Song, Panpan Zhu, Wanli Zhao, Tongyu Liu, Qi Wang, Tianming Zhao

**Affiliations:** 1College of Sciences, Northeastern University, Shenyang 110819, China; xuhaowen@stumail.neu.edu.cn (H.X.); 2000172@stu.neu.edu.cn (Y.S.); zhupanpan2004@163.com (P.Z.); 2Science and Technology on Electro-Optical Information Security Control Laboratory, Tianjin 300308, China; wanlizhao@163.com (W.Z.); liu_tongyu@163.com (T.L.)

**Keywords:** SPR, ZnO nanoflowers/Au structure, alcohol sensor, photochemical sensor

## Abstract

Alcohol detection plays a key role in food processing and monitoring. Therefore, we present a fast, high reproducibility and label-free characteristics alcohol photochemical sensor based on the surface plasmon resonance (SPR) effect. By growing ZnO nanoflowers on Au film, the SPR signal red-shifted in the visible region as the alcohol concentration increased. More interestingly, the sensitivity improved to 127 nm/%, which is attributed to the ZnO nanoflowers/Au structure. The goodness of the linear fit was more than 0.99 at a range from 0 vol% to 95 vol% which ensures detection resolution. Finally, a practical application for distinguishing five kinds of alcoholic drinks has been demonstrated. The excellent sensing characteristics also indicate the potential of the device for applications in the direction of food processing and monitoring, and the simple structure fabrication and economic environmental protection make it more attractive.

## 1. Introduction

Surface plasmon resonance (SPR) occurs when a photon of incident light hits a metal surface (typically a gold surface) [1,2,3,4]. At a certain angle of incidence, a portion of the light energy couples through the metal coating with the electrons in the metal surface layer, which then move due to excitation. The electron movements are now called plasmon, and they propagate parallel to the metal surface. Because of its good local field enhancement properties [5,6] and its ability to detect changes in refractive index (RI) through intermolecular interactions, SPR is regarded as a universal detection technique with good prospects in many bio-detection and sensing applications which is low cost, high reproducibility and label-free characteristics [7,8,9,10,11,12,13]. Therefore, the SPR technique has frequently been used for high-quality sensors.

Au film is commonly used for preparing SPR sensors due to the unusual optical properties, generating strong SPR signals at visible frequencies [14]. The SPR sensors based on Au are generally limited by their lack of high selectivity. One of the solutions is to modify sensing materials on the surface of Au film. ZnO has been widely investigated as a kind of sensing material due to its excellent properties with rich oxygen vacancies and low preparation cost [15,16]. The abundant vacancies on the surface of ZnO can adsorb molecules existing in the environment, such as H_2_O and alcohol, having an effect on the electrical conductivity of ZnO [17,18,19,20,21,22]. Therefore, modifying ZnO on the Au film can achieve a photochemical sensor for detecting alcohol concentration.

In this paper, a photochemical sensor for detecting alcohol concentration based on SPR has been realized. By growing ZnO nanoflowers on the Au film, the vacancies on the surface of ZnO will adsorb a large number of alcohol molecules. When the SPR phenomenon occurs, a large number of alcohol molecules will gain charge from the Au film, enhancing the SPR intensity and improving the sensitivity. The detection sensitivity is 127 nm/%, much higher than the sensor with Au films alone. Finally, the device is used for detecting the alcohol concentrations of five drinks. The excellent sensing characteristics show the potential application in food processing and monitoring at room temperature, and the simple and economical manufacture makes it attractive for a wider range of applications.

## 2. Materials and Methods

### 2.1. Materials

Zinc nitrate hexahydrate (Zn(NO_3_)_2_·6H_2_O) was purchased from Macklin Inc. (Shanghai, China), ammonia solution (25~28% NH_3_∙H_2_O), anhydrous ethanol (99%), and acetone (99.5%) were supplied by Sinopharm Chemical Reagent Co. (Tianjin, China). Au target material (99.99%) was purchased from Shenyang Kejing Auto-instrument Co. (Shenyang, China). All chemicals were analytic pure and utilized without further purification.

### 2.2. Synthesis of ZnO Nanoflowers/Au

The ZnO nanoflowers/Au structure on the prism was synthesized in two steps as shown in Figure 1. In the first step, the Au film was coated on the surface of the semi-cylindrical prism. First, the semi-cylindrical prisms (3 cm × 0.5 cm × 2 cm) are ultrasonic cleaning with a mixture of acetone and deionized water for 30 min and then dried in an oven. Afterwards, the cleaned semi-cylindrical prisms were coated with Au (50 nm) using a magnetron sputter coater. The sputtering conditions were set to the vacuum < 1 × 10^−2^ Pa, current ~10 mA, and vacuum evaporation time for 3 min. Finally, the device was dried in an oven for 2 h to enhance the adhesion.

In the second step, ZnO nanoflowers were synthesized on the Au film. The above device was placed in a solution containing 38 mL of deionized water, 0.8 g of zinc nitrate hexahydrate, and 2.3 mL of ammonia at 90 °C for 30 min [23,24,25]. Then the surface was cleaned with water several times to remove residual reagents. Finally, the photochemical alcohol sensor was dried with nitrogen gas.

### 2.3. Device Fabrication

A container (2.5 cm × 0.5 cm × 2 cm) was printed by 3d printer to hold liquid to be measured. The solution to be measured was prepared with a volume fraction.

### 2.4. Characterization and Measurement

The microstructure of the material was examined by scanning electron microscopy (SEM, Hitachi S4800, Tokyo, Japan). The crystalline phase of the material was characterized by X-ray diffraction (XRD, Dmax 2550 V, Cu Kα radiation, Osaka, Japan).

The measurement system consisted of a light source (Fiber optics tungsten halogen lamp, RLE-CP, Beijing, China), polarizer, diaphragm, rotatable carrier table, and spectrometer (Fiber spectrometer, RLE-SA03, Beijing, China). The spectrometer was used to measure the reflection intensity. The data received by the alcohol spectrometer at different concentrations was recorded by a computer. The tested temperature is ~5 °C.

## 3. Results

SPR is a surface electromagnetic mode at the metal-dielectric interface and is often used for detecting the changes in RI [26]. However, the only detecting RI of substances is hardly sufficient for today’s needs. In this paper, a ZnO nanoflowers/Au structure is designed to enhance the sensitivity to alcohol concentration. The ZnO nanoflowers/Au structures’ growth on semi-cylindrical prisms is shown in Figure 2a. ZnO is an inexpensive and nontoxic semiconductor metal oxide with abundant oxygen vacancies on its surface and it is often used as an alcohol sensing material [27]. The Au film has good SPR properties. Therefore, the structure can enhance the alcohol sensing properties. Figure 2b shows an optical graph of the device. Figure 2c shows the XRD pattern (Blue) and standard card (PDF#99-0111) of the ZnO nanoflowers. Peaks appearing at 31.7°, 34.4°, 36.2°, 47.5°, 56.5°, 62.8°, 66.3°, 67.9°, 69.1°, 72.5°, 76.9°, 81.3° and 89.6° corresponding to the (100) (002) (101) (102) (110) (103) (200) (112) (201) (004) (202) (104) planes can be indexed to ZnO crystal, respectively. Figure 2d,e shows a top-view SEM image and side-view SEM image of ZnO nanoflowers/Au structure, respectively. The thickness of Au film can be seen in Figure 2f and the thickness is ~50 nm. The short reaction time (30 min) is to control the density of ZnO nanoflowers and the length of ZnO, which can ensure the interactions of the SPR effect.

To determine the elemental composition, the energy-dispersive spectrometry (EDS) and elemental mapping are shown in Figure 3. Figure 3a shows the EDS image of the surface of the device. Figure 3b shows the enlarged view of Figure 3a. The peaks of Zn, O, and Au are attributed to the ZnO nanoflowers/Au structure. The elemental mapping of ZnO nanoflowers/Au structure is shown in Figure 3c–e, indicating that the materials contain ZnO and Au elements.

The tests of the contact angle between water and alcohol against Au film and ZnO nanoflowers/Au structure are shown in Figure 4. Figure 4a shows the water contact angle of Au film and the water contact angle is 86.9°. Figure 4b shows the water contact angle of the ZnO nanoflowers/Au structure and the water contact angle is 10.8°. Figure 4c shows the alcohol contact angle of Au film and the alcohol contact angle is 7.7°. Figure 4d shows the alcohol contact angle of the ZnO nanoflowers/Au structure and the alcohol contact angle is 1.3°. The water contact angle is far larger than the previous report [28]. This result may be due to the C contamination during the sputtering/storage process and surface microstructure by the sputtering process. Unfortunately, C contamination is inevitable in daily use. But by modifying ZnO nanoflowers can effectively lower the influences on physical adsorption on the material surface. Under the same exposure time in air, the water contact angle and alcohol contact angle of the ZnO nanoflowers/Au structure are both far smaller than that of Au film. The great affinity of the ZnO nanoflowers/Au structure may adsorb more alcohol molecules. These results suggest that the SPR sensors by nanoflowers/Au have a long-time life.

Figure 5a shows the SPR signal against different alcohol concentrations. Appendix A shows the relationship between refraction and various concentration of alcohol. The tested temperature is ~5 °C. With the increasing alcohol concentrations, the absorption valley of the device with Au film appears to red shifting. Figure 5b shows the fit curves between the position of absorption and alcohol concentration. The alcohol sensitivity (*S*) is defined by the equation:(1)$$S=\frac{\Delta \lambda }{\Delta C},$$
where the *λ* represents the position of the plasmon peak and the C represents the alcohol concentration (%). There is a linear fit at the range from 0 vol% to 95 vol%. The sensitivity is ~94 nm/% and the goodness of fit is 0.98. As shown in Figure 5c, the SPR signal also red shifts with the increasing alcohol concentrations. Interestingly, a linear fit at the range from 0 vol% to 95 vol% (Figure 5d). The sensitivity is ~127 nm/% and the goodness of fit is 0.99. In addition, compared to the SPR signal of Au film and ZnO nanoflowers/Au structure against 0 vol% alcohol concentration (Appendix A), the plasmon peak has an obvious red shift, which is attributed to the ZnO nanoflowers modification. As shown in Appendix A, the LOD of the ZnO nanoflowers/Au structure is 2%. The sensitivities of the two sensors are lower than the previous reports, which may be due to the impurities on the device surface in our work [29]. The plasma cleaner was purchased on the way, and we will solve this problem in our follow-up work.

## 4. Discussion

Figure 6 shows the reaction mechanism [30]. The visible light of the SPR takes place in the interface between alcohol solution and ZnO nanoflowers/Au through the prism (quartz) [31,32]. In addition, when the visible light propagates the interface, SPR will be excited, a matching condition expressed by the following equation [33]:(2)$$k_{0}n_{inc}\sin\theta_{inc}\mp m.\frac{2\pi }{\Lambda }=k_{spp}=k_{0}\sqrt{\frac{R_{e}\left(\varepsilon_{1}\right).\varepsilon_{2}}{R_{e}\left(\varepsilon_{1}\right)+\varepsilon_{2}}},$$
where *k*_0_ is the free-space propagation wavelength, *n_inc_* is the refractive index of the incident medium, m is the diffracted grating order, and *θ* is the incident angle, Λ is the grating period, *ε*_1_ is the complex permittivity of the metal and *ε*_2_ is the real permittivity of the surrounding dielectrics, and k*_SPP_* is the wave vector of the excited SPP mode. Therefore, the tiny changes in RI can be detected by our proposed device. The position of the resonance peak (*λ*) is satisfied with the following Equation (3):(3)$$\lambda \propto \frac{1}{\omega_{\gamma }}\propto \sqrt{1+2\varepsilon_{m}\;},\;$$
where *ω_γ_* indicates the resonance frequency and *ε_m_* is the dielectric constant of surrounding medium. When the RI increases, the red-shift of the SPR resonance wavelength can be observed (Figure 5).

When the device is exposed to the alcohol solution as shown in Figure 6a, the alcohol molecules are adsorbed on the surface of the Au film [34,35], combined with the electrons on the surface of the Au film, and rapidly dehydrogenated to produce H^+^ and C_2_H_5_OH^−^. When the SPR is exited, the H^+^ ion will be incident to trap the electrons of Au film and scatter faster by spillover mechanism [36,37,38,39]. Moreover, with the decreasing charge density of Au, photogenerated electrons of ZnO nanoflowers may flow to Au film, which enhances the SPR effect [40]. The possible response mechanisms can be represented by the following Equations (4) and (5):(4)$$C_{2}H_{5}\mathrm{OH}=C_{2}H_{5}O^{-}+H^{+},$$
(5)$$H^{+}+e^{-}=\;H,$$

Figure 6b shows the results of the COMSOL finite element analysis software. Against different concentrations of alcohol solution, the device with ZnO nanoflowers/Au structure has an outstanding performance. With the increase of the alcohol solution concentrations, the resonance peak is red-shifted. The results match our experiments. The relationship between concentrations of alcohol solution and resonance wavelength shows a good linear relationship. Therefore, our proposed device is expected to be an alcohol sensor for food detection. The linear fit of the simulation result is shown in Figure 6c. The sensitivity is ~66 nm/% and the goodness of fit is 0.98. However, the result is lower than that in the experiment. The chemical adsorption process (Figure 6a) enhances the surface local solution refractive index of ZnO nanoflowers/Au and lowers the conductivity of ZnO nanoflowers/Au. The effects together may lead to the result that the sensitivity in the experiment is larger than that in simulation.

To compare the performance of alcohol with ZnO nanoflowers/Au structures, the sensing performance for formaldehyde and formamide is shown in Figure 7. As shown in Figure 7a,c, various formaldehyde concentrations are measured by Au film and ZnO nanoflowers/Au structures, respectively. The resonance peaks are both red-shifted with the increasing formaldehyde concentrations. Figure 7b,d shows the linear fits of the relationship between formaldehyde concentrations and resonance peaks. However, the sensitivities are similar. Figure 7e–h shows the formamide sensing performance of two types of SPR sensors. The resonance peaks are both red-shifted with the increasing formamide concentrations while the sensitivities are no obvious differences. Appendix A show the relationship between the refractive index and concentration of formaldehyde and formamide at 5 °C, respectively. The refractive index increases with the concentration and the sensitivities to the refractive index are almost the same. Compared with the results in Figure 5, by modifying with ZnO nanoflowers, the sensors are more sensitive to alcohol than for formaldehyde and formamide, which is attributed to the chemical adsorption process (Figure 6a). Detailed data are shown in Table 1. The sensitivity of formaldehyde and formamide are lower than alcohol. Therefore, the alcohol sensor has a good application prospect in daily life.

The comparisons with other sensors have been shown in Table 2. The SPR sensors based on hollow-core fiber structures exhibit a high sensitivity of refractive index with 4125 nm/RIU [41]. However, this kind of SPR sensor has no performance improvement. The SPR sensors based on fiber Bragg grating also have no performance improvement, and the sensitivity is lower than that in this work [42]. These results suggest that the SPR sensors designed in this work demonstrate a potential application in food processing.

Finally, an application in detecting different alcohol drinks has been demonstrated. Five kinds of alcoholic drinks with different alcohol concentrations were purchased from a store and the alcohol concentration was achieved from nutrition information (Snowflake Pure Draft Beer: 2.5 vol%, Delirium Nocturnum: 8.5 vol%, Heiwen Shochu: 19 vol%, Original Sorghum: 38 vol%, Niulanshan Erguotou: 56 vol%). The SPR signals against different drinks are shown in Figure 8a. According to the linear fit, the results are shown in Figure 8b. The blue line represents the test results and the red line represents the standard value. It can be clearly seen that the test results are similar to the standard value, which exhibits great potential in alcohol sensing. In future applications, this technology will be used to accurately measure the amount of alcohol for food processing and monitoring and medical instrumentation.

## 5. Conclusions

In this paper, a highly sensitive photochemical alcohol sensor was achieved from the ZnO nanoflowers/Au structure. Due to the abundant vacancies on the surface of ZnO, alcohol molecules were easily absorbed, enhancing the SPR effect. The sensitivity to alcohol improved to 127 nm/% at an alcohol concentration range from 0 vol% to 95 vol%. The simple fabrication of the structure and its economic and environmentally friendly nature made it more attractive.

## Figures and Tables

**Figure 1 materials-15-00189-f001:**
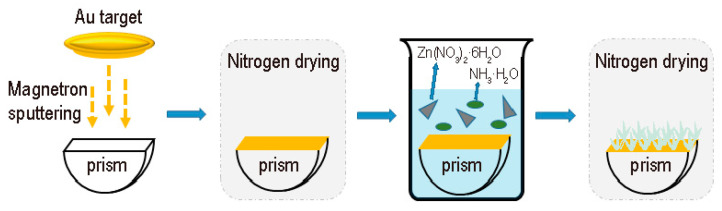
Schematic diagram of synthesis process.

**Figure 2 materials-15-00189-f002:**
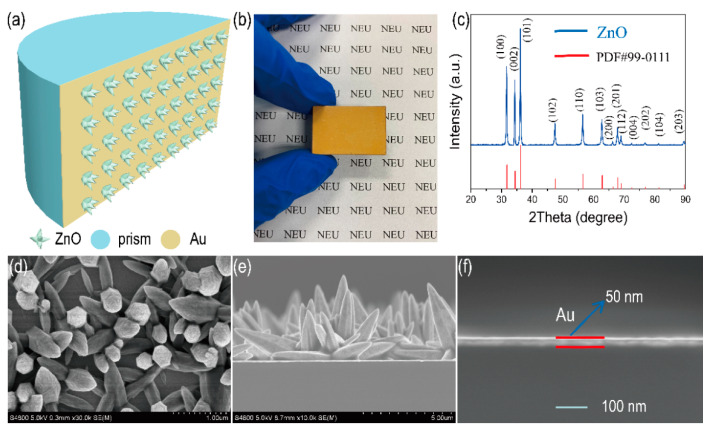
Schematic diagram of morphological characterization. (**a**) ZnO nanoflowers/Au structure. (**b**) Optical graph of the device. (**c**) XRD pattern of ZnO nanoflowers. (**d**) Top-view SEM image of ZnO nanoflowers/Au structure. (**e**) Side-view SEM image of ZnO nanoflowers/Au structure. (**f**) Side-view SEM image of Au film.

**Figure 3 materials-15-00189-f003:**
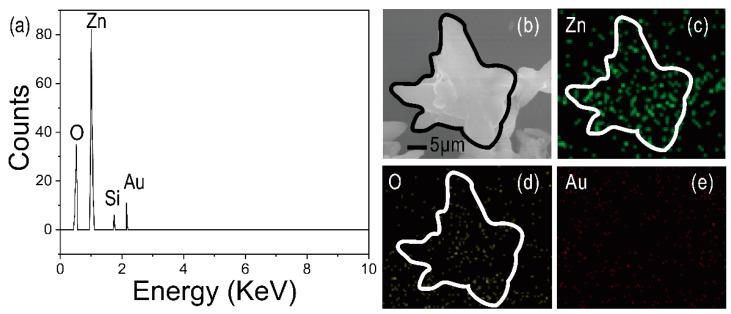
Elemental analysis. (**a**) EDS spectrum of ZnO nanoflowers/Au structure. (**b**) Enlarged view of ZnO nanoflowers/Au structure. (**c**) Elemental mapping of ZnO nanoflowers/Au structure: Zn, O (**d**) and Au (**e**).

**Figure 4 materials-15-00189-f004:**
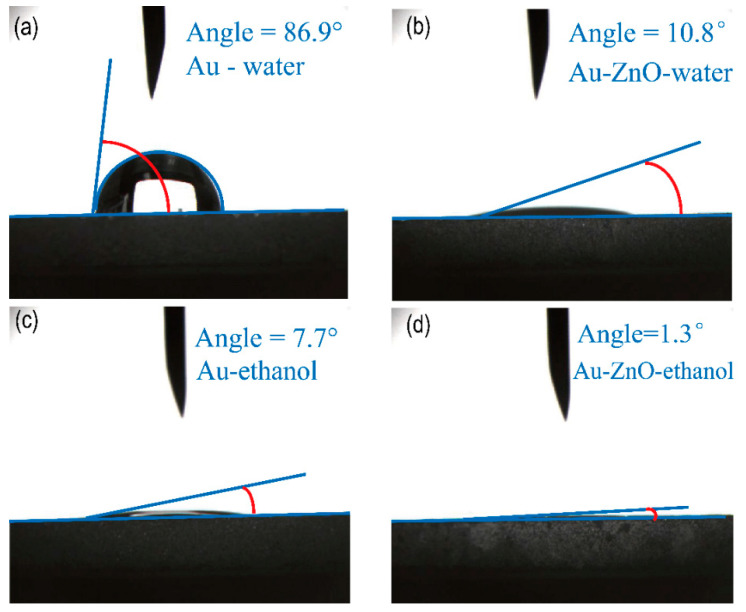
Schematic diagram of contact angle test. (**a**) Water contact angle of Au film (**b**) Water contact angle of ZnO nanoflowers/Au structures (**c**) Alcohol contact angle of Au film. (**d**) Alcohol contact angle of ZnO nanoflowers/Au structures.

**Figure 5 materials-15-00189-f005:**
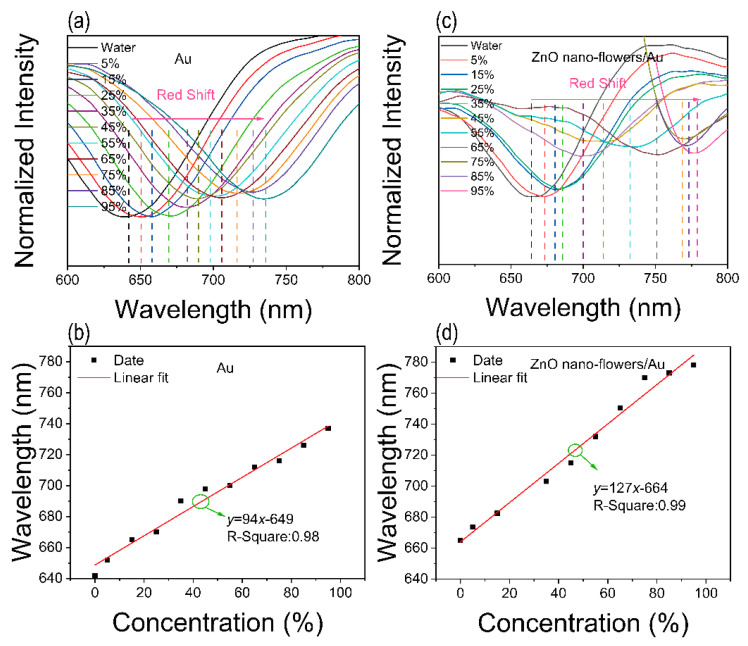
Schematic diagram of alcohol sensing performance. (**a**) SPR signal of the device with Au film against different alcohol concentrations. (**b**) Linear fit of the device with Au films against different alcohol concentrations. (**c**) SPR signal of the device with ZnO nanoflowers/Au structure against different alcohol concentrations. (**d**) Linear fit of the device with ZnO nanoflowers/Au structure against different alcohol concentrations.

**Figure 6 materials-15-00189-f006:**
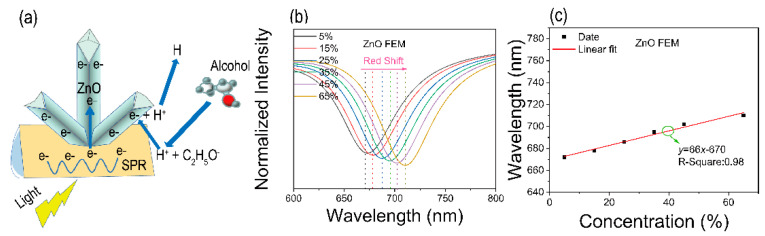
Schematic diagram of the principle. (**a**) Alcohol catalytic mechanism of ZnO nanoflowers/Au array structures. (**b**) COMSOL finite element simulation results. (**c**) The sensitivity calculated by COMSOL.

**Figure 7 materials-15-00189-f007:**
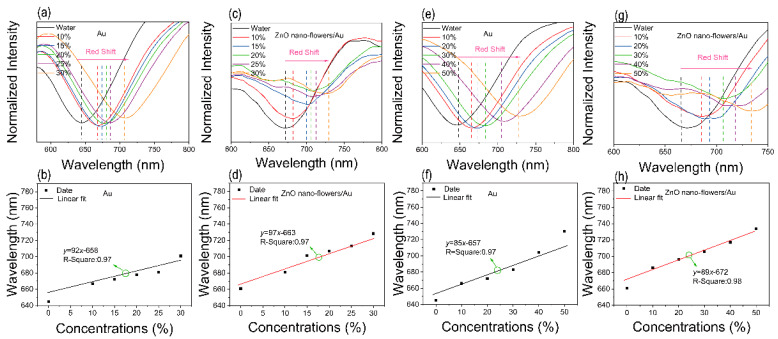
Schematic diagram of formaldehyde and formamide sensing performance. (**a**) SPR signal of the device with Au film against different formaldehyde concentrations. (**b**) Linear fit of the device with Au films against different formaldehyde concentrations. (**c**) SPR signal of the device with ZnO nanoflowers/Au structure against different formaldehyde concentrations. (**d**) Linear fit of the device with ZnO nanoflowers/Au structure against different formaldehyde concentration. (**e**) SPR signal of the device with Au film against different formamide concentrations. (**f**) Linear fit of the device with Au films against different formamide concentrations. (**g**) SPR signal of the device with ZnO nanoflowers/Au structure against different formamide concentrations. (**h**) Linear fit of the device with ZnO nanoflowers/Au structure against different formamide concentrations.

**Figure 8 materials-15-00189-f008:**
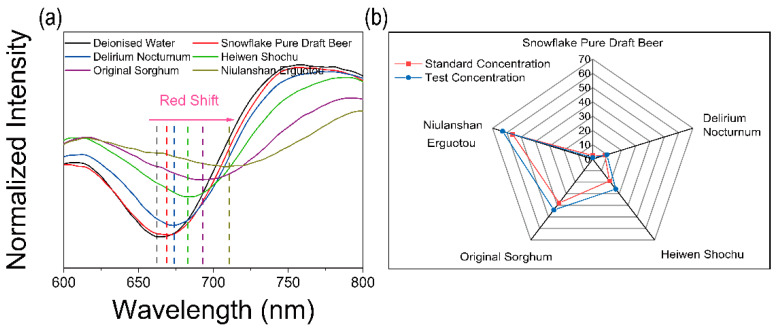
Practical application tests. (**a**) Alcohol sensing test. (**b**) Comparison of test concentration and standard concentration.

**Table 1 materials-15-00189-t001:** Comparison of the performance of three solutions.

Solutions of Detection	Range of Detection	Sensitivity to Concentration
Alcohol	0–95% (vol%)	127 nm/%
Formaldehyde	0–30% (vol%)	97 nm/%
Formamide	0–50% (vol%)	89 nm/%

**Table 2 materials-15-00189-t002:** Comparison of performance of three sensors.

Structure of Detection	Limit of Detection	Sensitivity to Concentration	Sensitivity for RI	Improving the Sensitivity for Alcohol	Reference
Hollow-core fiber	-	-	4125 nm/RIU	-	[40]
Fiber Bragg grating	-	-	500 nm/RIU	-	[41]
ZnO nanoflowers/Au	1.333 (2 vol/%)	127 nm/%	2825 nm/RIU	Alcohol	This work

## Data Availability

No new data were created or analyzed in this study. Data sharing is not applicable to this article.

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
