# Peer review of "Alcohol Sensor Based on Surface Plasmon Resonance of ZnO Nanoflowers/Au Structure"

_materials, 2021, doi:10.3390/ma15010189_

Round 1

Reviewer 1 Report

General comments:

The authors fabricated the alcohol detection sensor composed of ZnO nanoflowers grown on Au film on the prism. The sensitivity for the alcohol was enhanced two-fold by the addition of ZnO nanoflowers on Au film. Additionally, the practical application of the alcohol sensing of commercial beverages is meaningful.

However, in my opinion, a deeper scientific discussion is needed about spectral analysis and sensing mechanisms.

Therefore, I think minor revision is needed to bring this study to the level required for publication in Materials.

Specific Comments:

[1] The experimental condition of the alcohol sensing is unclear. If the authors use the “aqueous solution” of ethanol, they should describe it clearly. Additionally, the unit of concentration is not shown. Do they use whether wt% or vol%? Furthermore, what is measured for the sensing? If they measured the reflection, what do the y-axes mean in Figure 5a and 5b? (relative intensity to what? or normalized reflection?)

[2] Figure 5: How do the authors obtain the intercept value such as -1.5 and -3.1? From the calibration plots, the intercepts seem to be several hundred nm. Additionally, why don’t the authors show the spectrum of without alcohol? In the application part, the authors check the 2.5% alcohol concentration, which is outside of the calibration line.

[3] The authors compare experiment and simulation qualitatively, that is, both redshift according to the RI increment. How about the quantitative difference between them? (For example, the comparison of the sensitivity) In my understanding, FEM can only calculate the electromagnetic effect of the RI changes on SPR, which is expected from Equations (1) and (2), and the FEM does not consider the chemical effect explained in Fig. 6a. Therefore, If the ZnO-Au system dramatically enhances alcohol detection selectively, the sensitivity in the experiment should be larger than that in simulation.

[4] There are several simple errors in the manuscript. For example, Fig. 3a is the spectrum rather than the image. Also, the SI prefix of kilo should be a small letter. Also, εm is used without definition.

Author Response

Dear reviewer:

We are very grateful to the reviewers for their suggestions and comments on our manuscript (materials-1502556). These suggestions and comments are very helpful for us to improve the quality of our manuscript. We have carefully revised our manuscript according to the reports. Please see the attachement.

Thanks a lot for your time. We are looking forward to hearing from you.

Tianming Zhao

College of Sciences, Northeastern University, Shenyang 110819, People’s Republic of China

Tel: 86-18304011006

Email: zhaotm@stumail.neu.edu.cn

December 13, 2021

Reviewer 2 Report

The paper presents an SPR sensor based on Au+ZnO nanoflowers as a sensing structure. The demonstrated application is alcohol sensing. Although the topic would be interesting, the paper is not written well and also has many serious flaws. I cannot recommend its publication.

Major concerns:

  • There are no cleaning methods/protocols mentioned in the paper and based on the presented data, it seems to me that the authors measured on a not properly clean gold substrate. The first indication for this is the contact angle measurements. It is common knowledge that a clean gold surface is hydrophilic and should have contact angles well below 5o (org/10.1016/0021-9797(80)90348-3). The authors measured the initial contact angle on gold in water as 86.9o, which indicates serious carbon contamination. Thus, the contact angle measurements and their conclusions cannot be considered reliable.
  • The second indication is the SPR response. Based on the SEM images in Fig. 2 the size of the “nanoflowers” is in the um-5um range (considering their height). The size of these ZnO structures is definitely larger than the evanescent field penetration depth of the gold, which is ~200nm. Thus, the majority of the gold near-field is filled with ZnO. Since ZnO has a much higher refractive index than air or water, the gold should have a definite plasmon resonance peak shift upon deposition of the ZnO flowers. Yet, the position of the plasmon peak is at nearly the same position for 5% alcohol solution in Fig. 5. The ZnO-covered surface has the peak at an even shorter wavelength compared to “clean gold”. That simply cannot be.
  • For these two reasons, I find the authors' data unreliable and the conclusions and proposed mechanism for plasmonic sensing enhancement not properly supported.
  • Details of the Comsol FEM model and simulation are completely missing. I wonder how the authors implemented the proposed charge transfer mechanism and improved SPR sensing performance on the differential equation level in Comsol. Or is this only the response of the sensor due to refractive index changes?
  • The language of the paper is poor. Many sentences are hard to comprehend or formulated in an improper way. E.g. “However, overly chemical stability of Au film can hardly sense specific material.” “With the increase of the concentrations of alcohol solution, the resonance wavelength is red-shift.”
  • The selectivity of the proposed sensor towards alcohol is not demonstrated in any way.
  • The performance of the sensors should be expressed in the function of the RI changes in the medium. Sensitivity: nm/RIU (peak shift per unit RI). The LOD should be given for the alcohol concentration.
  • The performance of the proposed sensor should be compared with other solutions (plasmonic sensors) for alcohol sensing, in order to be able to assess its performance (sensitivity, LOD), advantages, disadvantages. This is completely missing.
  • The description of the SPR phenomenon is vague. It is not just a “surface electromagnetic mode localized” on the surface, but an oscillation in the electron density.

Author Response

Dear reviewer:

We are very grateful to the reviewers for their suggestions and comments on our manuscript (materials-1502556). These suggestions and comments are very helpful for us to improve the quality of our manuscript. We have carefully revised our manuscript according to the reports. Please see the attachment.

Thanks a lot for your time. We are looking forward to hearing from you.

Tianming Zhao

College of Sciences, Northeastern University, Shenyang 110819, People’s Republic of China

Tel: 86-18304011006

Email: zhaotm@stumail.neu.edu.cn

December 13, 2021

Round 2

Reviewer 2 Report

I reckon that the authors made much effort to correct their paper based on the comments. I really respect their hard work. However, there are still many mistakes, fundamental analytical errors, that cannot be neglected:

  • Before my earlier comment, every spectrum of the ZnO nanoflower modified electrode started from the same position as for the bare Au, from below 650 nm. Since my criticism, every spectrum corresponding to these nanoflowers miraculously shifted with 20 nm to comply with my comment. Except for the practical application test (fig. 8) and the simulations (fig 6) that still show the original spectra, starting from below 650 nm. Can the authors please explain how is this possible?
  • Even the spectra corresponding to bare Au are changed. What is more, it shows a non-linear behavior. Can the authors explain how is this possible in Fig. 5b? The bare gold shows a linear response in a larger range on other Figures e.g. 650-740 nm in Fig. 7f, why would it saturate for ethanol? Why was it different in the previous version of the paper?
  • The authors comment about the increasing contact angle upon storage is natural, due to contaminations. However, that is why a proper cleaning method should be implemented before any sensing measurements. Eg. chemical, or O2 plasma cleaning. Impurities, contamination will cause a drop in the SPR physical sensitivity…
  • …which is very transparent in the authors’ data. The normal RI sensitivity of a thin-film-based SPR should be well above 3000 nm/RIU. See: https://doi.org/10.1021/nl902721z The authors' sensitivity for pure gold is well below that for alcohol.
  • I have to stress that this is physical RI sensitivity, which should be the same for a given sensor if tested by changing the bulk RI of a medium. E.g. it should be the same for the authors (at least for pure gold) for the three tested media, ethanol formaldehyde and formamide. I can accept that using the ZnO nanostructures increases the sensitivity, but with 2800 nm/RIU it is still not better than a normal thin film-based SPR is, if prepared/cleaned properly.
  • The authors confuse basic analytical concepts, as in Tables 1 and 2. The 0-95% (in Tabe 1) or the RI range (in Table 2) is not the limit of detection. https://goldbook.iupac.org/terms/view/L03540
  • The authors also confuse selectivity. Their sensor might be more sensitive to ethanol, but it is not selective or specific to it. https://goldbook.iupac.org/terms/view/S05564 Naming the method selective in Table 2 is misleading. As with other SPR methods, any change in the RI of the medium will cause a signal that interferes with ethanol detection.
  • "Sensitivity to concentration" should also have a measure ([nm/%] for example). But a more proper alternative would be [nm/M]. Also, LOD and dynamic range should be in [M] units. Please see a proper table for comparison in here for example: https://doi.org/10.1016/j.snb.2016.02.084
  • The English of the paper is still not proper. There are many wording mistakes. E.g. “the plasmon peak has an obviously red shift, which is contributed to the ZnO nanoflowers modification.” Contributed -> attributed. Obviously -> Obvious.

Round 3

Reviewer 2 Report

Again, I have to reckon the authors' hard work in correcting their manuscript. The quality of the paper improved significantly, although I still find the data and the explanation a bit curious. All measurements should be performed at the same conditions, and measuring in a sample of 5oC is not too realistic. Thank you for your work, I leave the decision to the editor.

This manuscript is a resubmission of an earlier submission. The following is a list of the peer review reports and author responses from that submission.